# Classification of Holograms with 3D-CNN

**DOI:** 10.3390/s22218366

**Published:** 2022-10-31

**Authors:** Dániel Terbe, László Orzó, Ákos Zarándy

**Affiliations:** Institute for Computer Science and Control, H-1111 Budapest, Hungary

**Keywords:** digital holography, CNN, 3D-CNN, neural networks, deep learning

## Abstract

A hologram, measured by using appropriate coherent illumination, records all substantial volumetric information of the measured sample. It is encoded in its interference patterns and, from these, the image of the sample objects can be reconstructed in different depths by using standard techniques of digital holography. We claim that a 2D convolutional network (CNN) cannot be efficient in decoding this volumetric information spread across the whole image as it inherently operates on local spatial features. Therefore, we propose a method, where we extract the volumetric information of the hologram by mapping it to a volume—using a standard wavefield propagation algorithm—and then feed it to a 3D-CNN-based architecture. We apply this method to a challenging real-life classification problem and compare its performance with an equivalent 2D-CNN counterpart. Furthermore, we inspect the robustness of the methods to slightly defocused inputs and find that the 3D method is inherently more robust in such cases. Additionally, we introduce a hologram-specific augmentation technique, called hologram defocus augmentation, that improves the performance of both methods for slightly defocused inputs. The proposed 3D-model outperforms the standard 2D method in classification accuracy both for in-focus and defocused input samples. Our results confirm and support our fundamental hypothesis that a 2D-CNN-based architecture is limited in the extraction of volumetric information globally encoded in the reconstructed hologram image.

## 1. Introduction

In an in-line holographic microscope setup, the coherent light illuminates a dilute sample. Thus a large part of the light travels freely through the medium, which therefore acts as a reference beam, while the other part is diffracted by the objects within the sample. A camera sensor records the interference pattern of the reference and the diffracted waves, which results in the (raw) hologram image. This captured in-line hologram—unlike a conventional image—records all the volumetric information about the sample. Note that a CCD sensor can only record the intensity of the light; thus, the phase information is missing. Therefore, the numerically reconstructed image of this hologram at a special distance—called the reconstructed hologram—is biased by these missing, overlapping terms (e.g. twin image). Although there are phase recovery algorithms, these are iterative, computationally expensive procedures.

Several deep learning methods have been introduced recently in the fields of coherent imaging and digital holographic microscopy (DHM), showing superior performance to their conventional algorithmic counterparts. To name a few, it was applied to phase recovery and hologram reconstruction [1,2,3,4,5], to phase-unwrapping [6,7], to label-free sensing [8,9,10], to super-resolution [11], to object focusing [12], and even to make the transformation between coherent and incoherent imaging domains [13]. In our last work [14] we showed that the transformation between the domains is possible even without a paired dataset. Object classification is a common problem in DHM, as a holographic microscope is peculiarly suitable to inspect and measure large volumes rapidly, and classify/count micro-sized objects in a sample at a high speed. In [15], the authors employ a lens-free digital in-line holographic microscope to classify cells labeled with molecular-specific microbeads using deep learning techniques and show that the classification is feasible using the measured raw holograms only, although with lower performance compared to the case when reconstructed holograms are used. Zhu et al. in [16,17] developed a deep learning-based holographic classification method that operates directly on raw holographic data to obtain quantitative information about microplastic pollutants. MacNeil et al. in [18] utilized an in-line holographic microscope and deep learning solutions to classify planktons using reconstructed holograms.

3D convolutional neural networks are mainly applied in the field of medical imaging [19,20] and autonomous driving [21]; to our best knowledge, they have not been employed for DHM classification tasks yet. The nearest field where 3D CNN was applied is near-field acoustical holography [22,23].

The design of the convolutional neural network (CNN) was strongly influenced by the human visual system and functions very well on natural images (with incoherent illumination). We argue that as our eyes are not accustomed to coherent lighting conditions and hologram images, likewise the CNN is not optimal for hologram processing. We hypothesize that a convolutional neural network cannot effectively take advantage of the globally encoded volumetric information present in a hologram as a CNN works on local features inherently by its design; thus, it is preferable to apply preprocessing steps that extract the depth information. This can be done by numerically propagating the single input hologram to multiple distances and then concatenating them along a new depth dimension resulting in a volume. Next, this volumetric data—extracted from the single hologram—can be processed with a 3D CNN. The information content of the volumetric input is not increased but it is more explicit and decompressed, and more suitable for a CNN architecture, we claim. To validate our hypothesis, we construct a database for a hologram object classification task. After this, we implement a 3D CNN method—which has a volumetric input—and the corresponding 2D CNN—which has a single hologram input—to compare their performance. The automatic hologram focusing algorithm is not always accurate, therefore, the robustness of the classifier to slightly defocused inputs is a crucial factor that is also investigated in this work. The main contributions of this study are as follows:We debate that a CNN is optimal for utilizing all information present in a hologram and support this presumption with the results.We show that extracting the depth information by reconstructing a volume and feeding it to a 3D-CNN-based architecture improves the classification accuracy compared to the 2D-CNN baseline which operates on a single reconstructed hologram.We show that our 3D-model is inherently more robust to slightly defocused samples.Finally, we propose a novel hologram augmentation technique—called hologram defocus augmentation—that improves the defocus tolerance of both methods.

## 2. Materials and Methods

In this study, we use an in-line holographic microscope system. Our setup measures background-subtracted holograms of samples in a flow-through cell. After the raw hologram is captured, the objects present in the volume are localized, separately focused, and then cropped. As the applied automatic focusing algorithm is not always accurate it is manually corrected in the process of creating the training database.

We inspected the performance of the deep neural networks (DNN) on a challenging real-life problem: the classification of conglomerated objects into number-of-objects categories (i.e. quantifying the objects). The chosen specimen is an alga, namely Phaeodactylum tricornutum (TRC), which tends to cluster together mainly in the power of two groups. Thus, we recorded samples with our holographic microscope system and constructed a training database containing 1, 2, 4, 8, and 16 TRC classes. Additionally, we extended the database with two more categories: (1) small particles (SP) that are not TRCs but consistently smaller objects, and (2) poor-quality images or unidentifiable objects (debris). The captured hologram objects were manually focused. An example from each class is shown in Figure 1A. Usually, the larger the conglomerate the less numerous its occurrence is, therefore the database is imbalanced. The sample distribution can be seen in Figure 1B. Class weighting is applied in the loss function to balance the learning and the uneven database during neural network training.

We should mention that grouping into powers of two is an approximation and there are occurrences in-between categories (e.g., 3, 5, 11 conglomerates) which are placed into the closest neighboring classes. The labeling of the data is challenging and might be imprecise because it is often not clear to which class the given hologram belongs.

The objects in the hologram are digitally focused/reconstructed with the angular spectrum method [24] and then cropped with a 128 × 128 window around their centers. The reconstruction results in a complex image (z=a+ib), for which absolute value ($abs\left(z\right)=\sqrt{a^{2}+b^{2}}$) and angle ($ang\left(z\right)=tan^{-1}\left(b/a\right)$) is calculated and stored as amplitude and phase images. Our database consists of such hologram amplitude and phase image pairs arranged in (C = 2, H = 128, W = 128) sized arrays, where C denotes the channel number, H denotes the height, and W denotes the width.

Our proposed 3D architecture is explained in Figure 2. and the construction of its input is illustrated in Figure 1C. We take a sample from the dataset and propagate it 6 steps forward and 6 steps backward and concatenate them along a new dimension resulting in a tensor of shape (C = 2, D = 13, H = 128, W = 128), where D stands for the depth and one step is 3.43 μm. The backward and forward propagation terms are used to denote the direction of the propagation. For example, as shown in Figure 1C, if we propagate with 3.43 μm that is a forward step and if we propagate with −3.43 μm that is a backward step. This volumetric input is fed into the 3D DNN, which comprises four 3D residual convolution blocks (ResConvBlock) and a fully connected head (DenseBlock), which computes the class probabilities. The ResConvBlock consists of two 3D convolutions with LeakyReLU activations and batch norms. There is a skip connection after the convolutions and before the MaxPool operation. To accomplish this, we can not simply add the input to the output features because they have different channel numbers. Therefore, the input is projected with a 1 × 1 × 1 kernel 3D convolution into an output-sized tensor so they can be added (During neural network training, the gradient can backpropagate through the skip branch more freely because of the much simpler operations). The MaxPool operation in this block halves the spatial dimension and leaves the depth unaffected. In the DenseBlock, the whole depth is max pooled and the spatial dimension is average pooled to a size of (H = 2, W = 2), then, this tensor is flattened into a vector. A dropout with a drop probability of 0.5 is used before the linear layers. A LeakyReLU activation function is applied after the first linear layer and a LogSoftMax after the second and final one. Thus, the network was trained with a negative log-likelihood loss (NLLLoss) function:(1)$$L=-\sum_{c=1}^{N}w_{c}y_{c}l_{c}$$
where *N* is the number of classes, $w_{c}$ denotes the weights for class balancing, $y_{c}$ is the binary label (takes on 1 for the correct class otherwise 0), $l_{c}$ is the output of the network. As mentioned before, the last layer applies a LogSoftMax operation:(2)$$l_{c}=\log\left(\frac{e^{z_{c}}}{\sum_{i}^{N}e^{z_{i}}}\right)$$
where $z_{c}$ is the network output before the last normalization layer, which is described in the equation above. Note that this is equivalent to using cross-entropy loss when softmax is applied in the output layer.

To be able to evaluate the surmised advantage of the 3D network we also built its 2D alternative for baseline. For this 2D DNN, the input is simply the single reconstructed hologram amplitude and phase, 2D convolutions are employed instead of 3D ones and, obviously, the depthwise max pool operation in the DenseBlock is omitted. All the same channel and feature numbers are applied for a fair comparison.

The Pytorch deep learning framework is used to construct and train the neural networks. We split our data into training and test sets (85:15). Furthermore, we split the training set into 5-folds and conduct 5-fold validation training, using 4 folds for training and leaving out 1 fold for validation. Thus, we will have 5 models (with distinct parametrization) trained on altered datasets so we can have statistics on the performance of the methods when evaluating the multiple corresponding models on the test set. Adam optimizer is used for model parameter tuning with 2 $\times \;10^{-4}$ learning rate and 0.01 weight decay. An early stop is utilized to stop the training when the validation is not improved in the last 10 epochs; thus, the learnings took place for approximately 80 epochs in our data. Standard augmentations were employed (rotate, horizontal flip, and vertical flip) to avoid overfitting and to achieve better generalization.

To evaluate and compare the performance of the methods, we calculate the average accuracy and NLLLoss for every 5 models—which are trained with different fold configurations—on the test set and then compute their mean and standard deviation. Additionally, the F1-score matrix—i.e. the harmonic mean of the row-wise and column-wise normed confusion matrix—is constructed which provides insights into class-level performance.

Besides that, we also inspected the robustness of the methods to slightly defocused inputs. For this, we randomly propagate the original in-focus test holograms in the range from −13.72 μm to 13.72 μm which will contain slightly defocused and in-focus sample images too. We presume that the 3D method provides superior performance in the case of defocused inputs because it maps a whole volume from a single hologram—that will contain the in-focus section too if the mapped depth is sufficiently large and the resolution is adequately fine.

Additionally, we introduce a kind of data augmentation technique specific to holograms, called hologram defocus augmentation: we randomly propagate the input holograms in a specified range. In our experiment, this continuous range was [−13.72, 13.72] μm where the center (zero) represents the in-focus (not augmented) original input hologram, see examples in Figure 3. With this augmentation, we can effectively expand and generalize our database, which will result in increased defocus tolerance. We trained both 2D and 3D-models with and without hologram defocus augmentation and tested each one with in-focus inputs and also with defocused inputs. In the case of the 3D-model, the defocus augmentation means that the initial sample—from which the 3D input is created—is altered, which may result in the in-focus section being shifted from the middle of the constructed volume.

## 3. Results

The accuracy and the negative log-likelihood error (NLLLoss) of the different methods on the test set are shown in Table 1. The 3D-model outperforms the 2D-model in each scenario which entails that both the classification of in-focus holograms and the classification of slightly defocused holograms benefit from the 3D method. One could argue that a single hologram contains the same information as the volume—that is mapped from it with the angular spectrum method—thus, a DNN should be able to extract the information anyway. Our results contradict this argument and this may be explained by the fact that a CNN is by design working with local features while the hologram encodes depth information globally; thus, a CNN can not effectively learn to decode this information. Therefore, it is beneficial to map a single hologram to a volume—extracting its depth information—and then feed that in a 3D-CNN-based model.

For better visualization of the results, a boxplot of the accuracy is provided in Figure 4A. First, if we inspect the results with respect to models trained without hologram defocus augmentation (the first two models in Figure 4A), the 3D-model outperforms the 2D-model in all cases by approximately 4% considering accuracy. With hologram defocus augmentation the robustness of the 2D-model improves for defocused inputs, which thus attains the accuracy of the other method. For in-focused samples, however, the 3D-model performs still better. Moreover, if we train the 3D-model with hologram defocus augmentation too, then it surpasses the 2D-model in each input case again.

The F1-score matrices regarding both methods are shown in Figure 4B,C for in-focus test inputs and models trained without hologram defocus augmentation. The 3D-model has a superior classification performance for almost every class. The largest improvement is in the 16TRC category, which may be explained by a large conglomerated sample that may have parts that drift out of focus; thus, mapping it into a volume is advantageous as it contains many cross-sections where different parts of the large object are in focus. Consequently, the proposed method is especially beneficial in the case of large, complex objects which could not fit in a single cross-sectional plane.

## 4. Discussion

The results support our initial hypothesis that the 2D-model can not effectively utilize the global information present in the reconstructed hologram image: it has to be explicitly extracted by volumetric input construction and fed to the 3D-model. The proposed method has particular relevance in the case of large objects, which can be distinguished based on their whole 3D structure (for example some pollens). The classification of such objects can be solved by proper application of the presented approach.

The improvements are obvious, but even the accuracy of the 3D-model (∼75%) is considered to be very low considering the state-of-the-art CNN classification results. This relatively low score originates from the hard classification problem and from the inaccuracies of the database (for example, as mentioned in the methods section, there exist samples in-between categories). Furthermore, it is often not clear, even for a human, which class the reconstructed object belongs to, especially for larger conglomerates (e.g., 4, 8). Nevertheless, it is still suitable for comparing different methods.

The disadvantage of the 3D-method is that it is computationally very expensive. The training time of the 2D NN was approximately 10 minutes, while it was approximately 1 h and 30 min for the 3D NN on an Nvidia Tesla V100 GPU. Furthermore, the inference speed of the 2D method was an order of magnitude faster than the 3D method. In addition, the inference time does not involve the hologram preprocessing, i.e., the construction of the volumetric input, which is also time-consuming. Its memory requirement is also more than doubled. In the case of real-world applications, one should carefully consider whether the amount of performance improvement is worth the increased computation demand. But there are many ways to make the method more efficient; for example, both computation time and classification performance might be improved if the objects were copped after the 3D input construction.

In order to utilize the global information present on the hologram without increasing the required computations, one could apply Fourier Neural Networks (FNN)—which is the future direction of our research. A hologram can be represented as a complex image (containing the amplitude and phase information naturally) that can be processed in the Fourier space by element-wise multiplication with learnable filter matrices which have a global receptive field. Chen et al. in [25] applied FNN for phase reconstruction tasks with promising results. This approach could be adopted for classification tasks too.

## 5. Conclusions

This study investigates a 3D-CNN-based method for hologram classification. We propose a method—comprised of a 3D convolutional neural network architecture and volumetric input construction—that systematically utilizes the depth information present in a reconstructed hologram, and outperforms the 2D alternative baseline method. We show this through a challenging real-life classification problem. Moreover, we examine the tolerance of the methods to slightly defocused inputs. We found that the 3D-model is more robust to defocused inputs—as was expected considering its inherent structure. Finally, we introduce a novel augmentation technique specific to holograms, called hologram defocus augmentation, which improves the classification performance of both methods in the case of slightly defocused inputs. Based on the results of this preliminary study, we confirm our initial hypothesis and conclude that a 2D-CNN-based architecture is limited in the extraction of global information present in a hologram image.

## Figures and Tables

**Figure 1 sensors-22-08366-f001:**
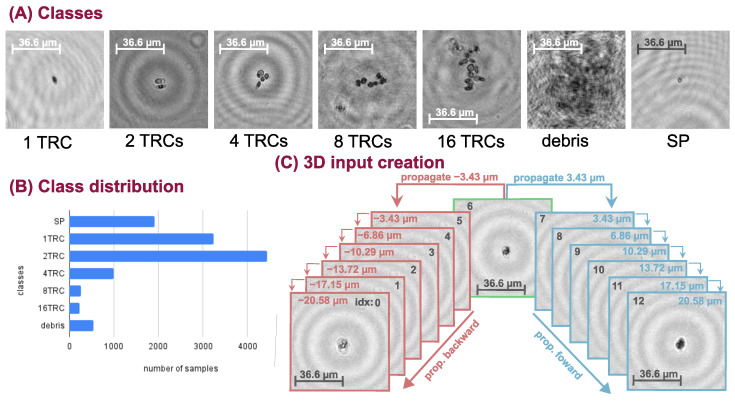
(**A**) Samples from the classes. (**B**) Distribution of the number of samples in classes. (**C**) The volumetric input creation for the 3D neural network. Note that only the hologram’s amplitude image is shown and the phase image is omitted in this illustration for the sake of simplicity. The backward and forward propagation terms denote the direction of the propagation.

**Figure 2 sensors-22-08366-f002:**
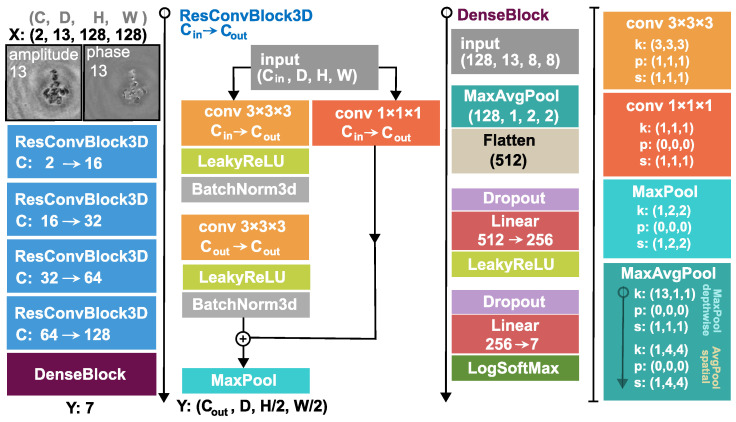
Illustration of the 3D-model architecture. The input of the 3D network is the amplitude and phase image (C = 2) of the initial hologram together with its 6 steps forward and 6 steps backward propagated forms (D = 13). The output of the network is the class log probabilities. The symbol *k* denotes the 3 dimensional kernel size (D,H,W), *p* the padding, and *s* the stride.

**Figure 3 sensors-22-08366-f003:**
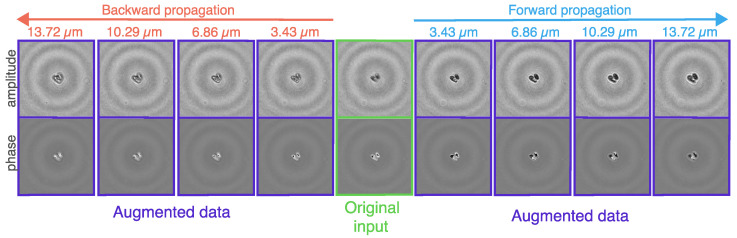
Examples of the application of hologram defocus augmentation for a sample in class 2-TRCs. In this illustration, from the original in-focus input hologram, we generate 8 slightly altered examples by propagating the hologram in the range of [−13.72, 13.72] μm. The backward and forward propagation terms denote the direction of the propagation. Backward propagation means that we propagate in the negative direction: for example, propagating one step backward is equivalent to propagating with −3.43 μm.

**Figure 4 sensors-22-08366-f004:**
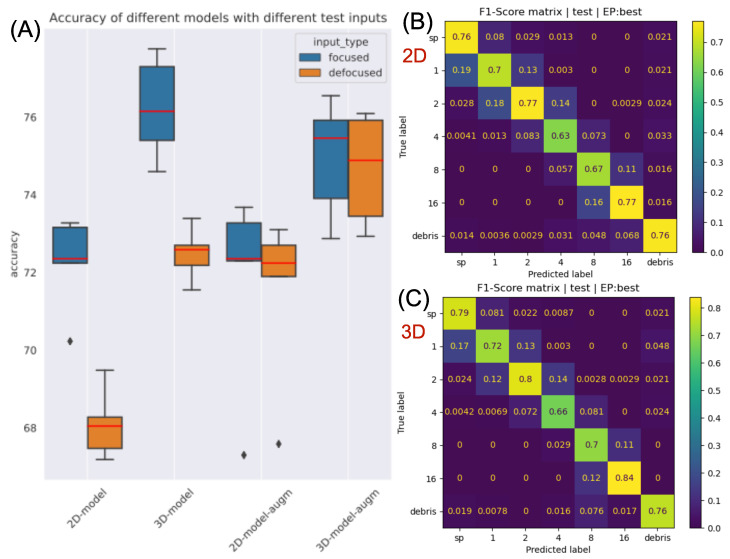
(**A**) Boxplot of the accuracy of the models with different input types; (**B**) F1-score matrix of the 2D-model in the case of in-focus training and test inputs. (**C**) F1-score matrix of the 3D-model in the case of in-focus training and test inputs.

**Table 1 sensors-22-08366-t001:** Performance of the different models. The models marked with *augm* are trained with hologram defocus augmentation. Top scores are highlighted in bold.

Model	Input Type	Accuracy	NLLLoss
2D-model	in focus	72.3±1.2	0.711±0.021
3D-model	in focus	76.2±1.3	0.621±0.019
2D-model-augm	in focus	71.78±2.6	0.721±0.02
3D-model-augm	in focus	74.9±1.5	0.655±0.044
2D-model	defocused	68.3±0.9	0.811±0.021
3D-model	defocused	72.5±0.7	0.702±0.015
2D-model-augm	defocused	71.5±2.2	0.732±0.05
3D-model-augm	defocused	74.7±1.4	0.657±0.043

## Data Availability

Data underlying the results presented in this paper are not publicly available (may be obtained from the authors).

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
