# Peer review of "Classification of Holograms with 3D-CNN"

_sensors, 2022, doi:10.3390/s22218366_

Round 1

Reviewer 1 Report

This paper proposes a classification method for the number of objects using a 3D-CNN technique. The authors compared the proposed method to a conventional 2D-CNN and showed that it is inherently more robust to out-of-focus samples than 2D-CNN. Although the classification accuracy did not improve significantly, the idea is novel and very interesting. Therefore, the reviewers believe that the paper is worthy of publication if more detailed explanations are added to the points listed below.

- Why does the 3D-model work well for 2D image input?

- What is the difference between 2D-model-augm and 3D-model?

- Does 3D-model-augm mean that the Depth region was further expanded and learned?

Author Response

- Why does the 3D-model work well for 2D image input?

The 3D-model operates on a volumetric input. The original 2D image is mapped into a 3D input as illustrated in Figure 1/C.

- What is the difference between 2D-model-augm and 3D-model?

2D-model-augm denotes that during the 2D model training the input samples were slightly defocused, resulting in increased performance for defocused object classification in test time. The 3D-model’s input is 12 cross-sectional planes of the object at different depths, so it has a volumetric input and is inherently more robust to defocus as one of the planes will be in focus. Moreover, large objects might not fit in one cross-sectional plane meaning that the 2D-model can observe just a part of the object, however, the 3D-model can scan through it.

- Does 3D-model-augm mean that the Depth region was further expanded and learned?

No, the depth region is fixed, only the starting point is changed. In other words, the initial sample is altered from which the 3D input is created. This will result in that the focused cross-section won’t be in the middle of the volume, it is shifted. This is a good question, it was not discussed in the article which is now corrected and added to the end of the materials section.

Reviewer 2 Report

- The results can be greatly improved if the images are cropped to include only the relevant information.

- What is meant by propogating the image backward and forward? 

- Only two performance metrics are included. Other metrics (e.g., precision) should be included. Also, define all metrics in the context of multiclass classification.  

Author Response

- The results can be greatly improved if the images are cropped to include only the relevant information.

In a hologram image, the rings-like diffraction pattern of the object may encode valuable information. Furthermore, it can be effectively propagated to different depths only if a sufficiently large surrounding is available. Nevertheless, it is an interesting suggestion and might be a topic of another study to inspect whether the 2D-model utilizes the ring pattern of the hologram object. This idea is now included in the revised manuscript in the discussion part.

- What is meant by propagating the image backward and forward?

A 2D hologram image encodes volumetric information in its interference pattern and we can numerically propagate in different depths. The backward and forward propagation terms are used to denote the direction of the propagation. For example, in Figure 1/C, if we propagate with 3.43 um that is a forward step and if we propagate with -3.43 um that is a backward step considering the applied focal position of the hologram reconstruction. This explanation is added to the materials section (after line 109) and also to the related Figure captions.

- Only two performance metrics are included. Other metrics (e.g., precision) should be included. Also, define all metrics in the context of multiclass classification.

Though in the table only two metrics are present, in Figure 4/B-C, F1-score matrices are included. The F1-score matrix is the harmonic mean of the sensitivity-and precision matrix which provides insight into the class-level performance of the models – which is important in a multiclass classification problem.

Reviewer 3 Report

A hologram, when measured with appropriate coherent illumination, records all significant volumetric information about the sample. It is encoded in its interference patterns, and from these, the image of the sample objects can be reconstructed in various depths using standard digital holography techniques. The authors claim that because a 2D convolutional network (CNN) operates on local spatial features, it cannot be efficient in decoding this volumetric information spread across the entire image. As a result, the authors propose a method in which we extract the hologram's volumetric information by mapping it to a volume - using a standard wavefield propagation algorithm - and then feeding it to a 3D-CNN-based architecture. This work need to address some address revisions/concerns before final publication.

1. What is novelty of the work. Please underscore the scientific value added/contributions of your paper in your abstract and introduction and address your debate shortly in the abstract.

2. A good article should include, (1) originality, new perspectives or insights; (2) international interest; and (3) relevance for governance, policy or practical perspective.

3. The work is devoted to an actual scientific and applied problem, performed by correct modern methods and the results are not in doubt. But the presentation and discussion of the results, as well as the conclusions, need to be improved.

4. Literature review section need to be enhanced. Add some recent related works. Some suggested works are :

Wang, J., Zhang, Z., Huang, Y., Li, Z., & Huang, Q. (2021). A 3D convolutional neural network based near-field acoustical holography method with sparse sampling rate on measuring surface. Measurement177, 109297.

Shi, L., Li, B., Kim, C., Kellnhofer, P., & Matusik, W. (2021). Towards real-time photorealistic 3D holography with deep neural networks. Nature591(7849), 234-239.

Wang, J., Zhang, Z., Li, Z., & Huang, Q. (2022). Research on joint training strategy for 3D convolutional neural network based near-field acoustical holography with optimized hyperparameters. Measurement202, 111790.

Shimobaba, T., Kakue, T., & Ito, T. (2018, June). Convolutional neural network-based regression for depth prediction in digital holography. In 2018 IEEE 27th International Symposium on Industrial Electronics (ISIE) (pp. 1323-1326). IEEE.

5.  What is backward and forward propagation in augmentation. Discuss.

6.  Write mathematical equation for NLLLoss.

7. Figure 4. (A) Boxplot of the accuracy of different models with different input types; F1-score matrix of the 2D-model (A) and 3D-model (B) in case of in-focus training and test inputs. What is C? are these confusion matrices?

Author Response

  1. What is the novelty of the work? Please underscore the scientific value added/contributions of your paper in your abstract and introduction and address your debate shortly in the abstract.

“We claim that a 2D convolutional network (CNN) cannot be efficient in decoding this volumetric information spread across the whole image as it inherently operates on local spatial features. Therefore, we propose a method, where we extract the volumetric 6 information of the hologram by mapping it to a volume – using a standard wavefield propagation algorithm – and then feed it to a 3D-CNN-based architecture.” as it is stated in the abstract. We provide some reasoning and some experimental demonstration/proof of these claims. 

We extended the abstract and now it contains a more detailed description of the results and conclusions.

  1. A good article should include, (1) originality, new perspectives or insights; (2) international interest; and (3) relevance for governance, policy, or practical perspective.

It has particular relevance in the case of large objects which can be differentiated based on their whole 3D structure (for example some pollens), the classification of such objects can be solved by proper application of the proposed method. The discussion part is extended with more information regarding the practical perspective.

  1. The work is devoted to an actual scientific and applied problem, performed by correct modern methods and the results are not in doubt. But the presentation and discussion of the results, as well as the conclusions, need to be improved.

The introduction, discussion, and conclusion parts are extended with more details.

  1. Literature review section need to be enhanced. Add some recent related works. Some suggested works are…

The bibliography is extended with some of the suggested and some other works.

  1. What is backward and forward propagation in augmentation? Discuss.

The backward and forward propagation terms denote the direction of the propagation. Backward propagation means that we propagate in the negative direction: for example, propagating one step backward is equivalent to propagating with -3.43 um. 

This information is added to the related Figure captions and further discussion is included in the materials section (after line 109).

  1. Write the mathematical equation for NLLLoss.

The mathematical formula and additional information regarding the loss are provided in the revised manuscript.

  1. Figure 4. (A) Boxplot of the accuracy of different models with different input types; F1-score matrix of the 2D-model (A) and 3D-model (B) in case of in-focus training and test inputs. What is C? are these confusion matrices?

There was a mistake in referencing the subfigures which is now corrected: (A) Boxplot of the accuracy of different models with different input types; F1-score matrix of the 2D-model (B) and 3D-model (C) in case of in-focus training and test inputs. (B-C) are F1-Score matrices which is the harmonic mean of the column-wise and row-wise normed confusion matrix (in other words the harmonic mean of the precision and sensitivity matrices).